# Evaluation of Dynamic Changes of Volatile Organic Components for Fishmeal during Storage by HS-SPME-GC-MS with PLS-DA

**DOI:** 10.3390/foods13091290

**Published:** 2024-04-23

**Authors:** Jie Geng, Qing Cao, Shanchen Jiang, Jixuan Huangfu, Weixia Wang, Zhiyou Niu

**Affiliations:** 1College of Engineering, Huazhong Agricultural University, Wuhan 430070, China; gengjie@webmail.hzau.edu.cn (J.G.); cqing2267464@outlook.com (Q.C.); jiangshanchen@webmail.hzau.edu.cn (S.J.); 15623004460@webmail.hzau.edu.cn (J.H.); 15296706307@163.com (W.W.); 2Key Laboratory of Smart Farming for Agricultural Animals, Ministry of Agriculture, Wuhan 430070, China

**Keywords:** fishmeal, HS-SPME-GC-MS, freshness, correlation analysis, PLS-DA

## Abstract

Headspace solid-phase microextraction, combined with gas chromatography–mass spectrometry and partial least squares discriminant analysis, was adopted to study the rule of change in volatile organic compounds (VOCs) for domestic and imported fishmeal during storage with different freshness grades. The results showed that 318 kinds of VOCs were detected in domestic fishmeal, while 194 VOCs were detected in imported fishmeal. The total relative content of VOCs increased with storage time, among which acids and nitrogen-containing compounds increased significantly, esters and ketones increased slightly, and phenolic and ether compounds were detected only in domestic fishmeal. Regarding the volatile base nitrogen, acid value, pH value, and mold counts as freshness indexes, the freshness indexes were significantly correlated with nine kinds of VOCs (*p* < 0.05) through the correlation analysis. Among them, volatile base nitrogen had a significant correlation with VOCs containing nitrogen, acid value with VOCs containing carboxyl group and hydrocarbons, pH value with acids which could be used to adjust pH value, and mold counts with part of acids adjusting pH value and VOCs containing nitrogen. Due to the fact that the value of all freshness indexes increased with freshness degradation during storage, based on volatile base nitrogen and acid value, the fishmeal was divided into three freshness grades, superior freshness, corrupting, and completely corrupted. By using partial least squares discriminant analysis, this study revealed the differences in flavor of the domestic and imported fishmeal during storage with different freshness grades, and it identified four common characteristic VOCs, namely *ethoxyquinoline*, *6,7,8,9-tetrahydro-3H-benzo[e]indole-1,2-dione*, *hexadecanoic acid*, and *heptadecane*, produced by the fishmeal samples during storage, as well as the characteristic VOCs of fishmeal at each freshness grade.

## 1. Introduction

Fishmeal is a nutrient-rich protein source derived from fish, known for its abundant supply of essential fatty acids, minerals, and vitamins. It possesses excellent digestibility and is highly palatable [1]. Due to these qualities, fishmeal is extensively utilized in the formulation of animal feed for livestock, poultry, and aquatic species. Additionally, aroma plays a pivotal role in determining the overall quality and consumer perception of fishmeal [2]. Studying the various aroma compounds would improve the understanding of fishmeal processing and broaden the methods of evaluating the freshness grades. Many factors, such as raw materials, processing technology, storage conditions, and freshness grades, would influence the quality of fishmeal [3,4]. At present, research has only been conducted on the freshness grades of fishmeal [3]. Moreover, the difference resulting from the processing technology of domestic and imported fishmeal would naturally lead to different aroma characteristics of fishmeal during storage with different freshness grades. Especially, characteristic volatile organic compounds (VOCs) in domestic and imported fishmeal during storage with different freshness grades need to be identified and compared. Nevertheless, past studies primarily emphasized the analysis and characterization of VOCs within a specific type of fishmeal, rather than exploring variations across multiple varieties. Currently, scientific information on VOCs and the rule of change for domestic and imported fishmeal during storage is limited. 

The advancement of instrumental analysis technology has led to the adoption of gas chromatography–mass spectrometry (GC-MS) for the identification and quantification of VOCs. GC-MS has found extensive application in food analysis and quality assessment, an essential part of which is sample extraction [5,6]. Headspace solid-phase microextraction (HS-SPME) is an effective pretreatment process for extraction and is commonly combined with GC-MS for detecting contents [7,8]. Due to its high sensitivity and good reproducibility, HS-SPME-GC-MS is widely used in fruits, grains, liquor, and meat [9,10,11,12]. Li et al. analyzed the mechanism of aroma formation during passion fruit ripening based on HS-SPME-GC-MS combined with transcriptome analysis [13]. Xi et al. determined that *1,8-cineole* and *ethanol* as the characteristic VOCs in the early ripening stage of walnut oil, *nonanal*, *(E)-2-octenol*, and *hexanal* in the mid-ripening stage; and *nonanal*, *1-heptanol*, *heptanal,* and some organic acids in the later ripening stage [14]. In recent years, more and more researchers have focused on HS-SPME-GC-MS to detect and quantify VOCs and combine them with freshness indexes to explore the correlation. Zhou et al. studied the correlation between odor and freshness of channel catfish during cold storage based on HS-SPME-GC-MS, indicating that *benzaldehyde* and *1-octene-3-alcohol* could be used for characterizing spoilage [15]. Zhang Jingjing found that the VOCs related to spoilage in *larimichthys polyactis* and *argyrosomus argentatus* were *trimethylamine*, *indole,* and *3-methyl-butanol* [16]. However, the connection between VOCs and the perception of freshness during storage of fishmeal is commonly made by few people. The above research studies have a reference meaning for determining VOCs that characterize the freshness grades of fishmeal.

In this study, the HS-SPME-GC-MS technique was employed to conduct a rapid analysis and detect VOCs of domestic and imported fishmeal during storage for 30 days. Combined with freshness indexes, including volatile base nitrogen (VBN), acid value (AV), pH value, and mold counts, tests for the significance of Pearson’s correlation coefficient were adopted to analyze the correlation between each freshness index and VOCs. Based on the analysis of the partial least squares discriminant analysis (PLS-DA) model, the differences in VOCs in domestic and imported fishmeal and with different freshness grades were clarified, and the characteristic VOCs were determined. This study aimed to find the VOCs as freshness markers and provide a theoretical basis for storage, quality control, and subsequent utilization of fishmeal.

## 2. Materials and Methods

### 2.1. Fishmeal Sample

In this study, fishmeal samples were used of freshly domestic defatted fishmeal and imported Peruvian steam-dried fishmeal, purchased from the Weihai Yongheng Seafood process plate and Technologica de Alimentos S.A. Company, respectively, whose product labels about composition are shown in Table 1. The production process of fishmeal is as follows: selecting raw materials, cooking, pressing, steam drying, cooling, screening, and crushing. Anchovy fish is the raw material of Peruvian fishmeal, while the raw material of domestic fishmeal is more complex with many kinds of fish. Due to the domestic defatted fishmeal, there is solid–liquid separation and oil–water separation in the production process between pressing and drying. The domestic fishmeal is usually packaged by direct filling. Considering transportation, the imported fishmeal is usually packaged in a packaging bag covered with plastic film after deodorization.

In total, 5 kg of fresh fishmeal laid on the enamel plate was stored in a constant-temperature and relative-humidity (RH) incubator at 25 °C 80%RH for 30 days, respectively (RGX-250B, Tianjin Sailidesi Experimental Analysis Instrument Factory, Tianjin, China). According to the quartering method based on the national standard GB/T 19164 [17], 200 g of fishmeal samples was taken every other day for testing. The number of samples analyzed was 16 for each type of fishmeal, a total of 32 samples.

### 2.2. Chemicals and Reagents

Anhydrous ethanol, anhydrous ether, potassium hydroxide (KOH), hydrochloric acid (HCl), sodium hydroxide (NaOH), boric acid (H_3_BO_3_), sodium chloride (NaCl), phenolphthalein, methyl-red dye, bromocresol green, and magnesium oxide (MgO) were acquired from the Sinopharm Chemical Reagent Co., Ltd. (Shanghai, China). *3-methyl-3-buten-1-ol* (used as an internal standard) was acquired from the Macklin Biochemical Technology Company (Shanghai, China). *Paraffin hydrocarbons (C_6_–C_23_)* were acquired from Sigma-Aldrich (St. Louis, MO, USA). The distilled water used was from Watsons (Watsons Food and Beverage Co., Ltd., Guangzhou, China). All the chemicals utilized in this study were analytical or chromatographic grade.

### 2.3. Detection Methods of Freshness Indexes

The volatile base nitrogen (VBN), acid value (AV), pH value, and mold counts were used as the freshness indexes of fishmeal. The determination of VBN referred to the automatic Kjeldahl nitrogen analyzer method based on the national standard GB/T 19164 [17]. The determination of AV referred to the titration neutralization method based on the national standard GB/T 19164 [17]. The pH value determination followed the guidelines in the national GB 5009.237 [18]. The mold counts were determined following the national standard GB/T 13092 [19].

### 2.4. Extraction of VOCs by HS-SPME

The VOCs were extracted by HS-SPME. According to the previous study, the optimized extraction conditions for fishmeal samples were as follows: 50/30 μm divinylbenzene/Carboxen^®^/polydimethylsiloxane (DVB/CAR/PDMS) fiber head, extraction temperature of 87 °C, equilibration time of 24 min, extraction time of 60 min, and the addition of saturated saline volume of 4 mL. Precisely weighted, 3 ± 0.001 g of fishmeal samples was added to a 20 mL headspace vial filled with silica gel, followed by the addition of 4 mL saturated saline volume and 100 μL *3-methyl-3-buten-1-ol* (1%) as an internal standard. And the headspace vial was sealed immediately. After equilibrating the fishmeal samples by water-bath heating (LC-MSB-HD, Lichen Company, Shanghai, China) at 87 °C for 24 min, the 50/30 μm DVB/CAR/PDMS fiber head installed in the SPME fiber holder (57330-U, Supelco Inc., Bellefonte, PA, USA) was then inserted into the vial in headspace for 60 min at 87 °C to extract the VOCs of fishmeal. After the completion of extraction, the SPME fiber was retrieved from the vial and inserted into the injection port of GC for desorption, lasting for 5 min at 260 °C. To prevent any potential contamination, the extracted fibers were required to be kept at 260 °C for 1 h in the injection system of GC-MS before usage.

### 2.5. VOCs Profiling via GC-MS

For this experiment, GC-MS (GC8890/7000D, Agilent Technologies, Inc., Santa Clara, CA, USA) and a quartz capillary column (HP-5MS UI, 30 m × 0.25 mm × 0.25 μm, Supelco Inc., Bellefonte, PA, USA) were utilized. The injection temperature was 260 °C. The carrier gas was high-quality helium (He, >99.999%) at a flow rate of 1 mL/min; the sample was injected by splitless. The column temperature was 40 °C initially and then increased to 140 °C at a rate of 4 °C/min, maintaining for 2 min. Finally, the temperature was increased to 260 at a rate of 10 °C/min, maintaining for 5 min. The transfer line temperature was maintained at 280 °C. The quadrupole mass spectrometer was operated in the electron impact (EI) mode, and the ion source temperature was set at 230 °C. Initially, full scan-mode data were acquired to determine appropriate masses in selected ion monitoring (SIM) mode under the scan range of m/z 30–500 amu. All analyses were performed with a setting electron ionization energy of 70 eV.

### 2.6. Identification of VOCs

The data were extracted from the database of the system by filtering for compounds exhibiting a similarity of over 70%. Retention indexes (RIs) were then determined utilizing a mixture of *paraffin hydrocarbons (C_6_–C_23_)*. After that, the VOCs were matched with the National Library of Standard and Technology (NIST 17. L) spectral library.

The relative content of each VOC was assessed according to the internal standard method that the ratio of peak areas is proportional to the concentration of VOCs. The relative content could be calculated as shown in Equation (1).
(1)$$C_{x}=C_{0}\times V_{0}\times S_{x}/\left(S_{0}\times m\right)$$
where *C_x_* represents the relative content of the unknown VOCs (μg/kg); *C*_0_ represents the concentration of the internal standard (μg/μL); *V*_0_ represents the addition of volume of the internal standard (μL); *S_x_* represents the peak area of the unknown VOCs (AU·min); *S*_0_ represents the peak area of the internal standard (AU·min); and *m* represents the mass of fishmeal samples (kg).

### 2.7. Statistics Analysis

Detections of freshness indexes and VOCs were carried out in at least three independent evaluations, expressed as the mean values ± standard deviation (SD). The software SPSS 24 (IBM Inc., Chicago, IL, USA) was used for data processing. The analyses of variance (ANOVAs) were performed among the means, using the least significant difference (LSD) to analyze the differences between VOCs (*p* < 0.05). The correlations between the freshness indexes (VBN, AV, pH value, and mold counts) and VOCs were calculated by Spearman correlation coefficients. The software Origin Pro 2023 (Origin Lab, Northampton, MA, USA) was used to plot. MetaboAnalyst 5.0 (https://www.metaboanalyst.ca/, accessed on 18 December 2023) was used for PLS-DA.

## 3. Results and Discussion

### 3.1. Regularity of Freshness Indexes during Storage

The changes in fishmeal freshness indexes during storage are shown in Figure 1. The VBN showed a trend of slow decrease–slow increase–significant increase. After 26 days of storage, the value of VBN exceeded 160 mg/100 g, and the fishmeal was completely spoiled (Figure 1a). This phenomenon occurred due to the enhanced enzymatic activity and accelerated decomposition rate of nitrogen-containing compounds under a higher RH. Consequently, the levels of ammonia, trimethylamine, and dimethylamine increased [20]. The AV showed an overall upward trend. Compared with imported fishmeal, the change in domestic fishmeal fluctuated greater (Figure 1b). Under higher RH storage, the reproduction and growth of microorganisms promoted the oxidative rancidity of fat. So, the AV continued to increase with storage time and the freshness grade of fishmeal was reduced. The initial rapid decline in pH value could be attributed to the liberation of inorganic phosphate resulting from the degradation of ATP and/or the accumulation of lactic acid during anaerobic glycolysis. Subsequently, the pH value may gradually increase, which could be attributed to the accumulation of alkaline compounds such as biogenic amines and ammonia, eventually reaching a stable state. Finally, after the completely spoiled, the pH value of fishmeal increased significantly [21,22] (Figure 1c). The process of reduction of freshness would be accompanied by the growth of molds. The reproduction of mold necessitates the consumption of nutrients and results in the production of toxic secondary metabolites known as mycotoxins. Thus, the mold counts were used as one of the freshness indexes. After the storage of 26 days, the mold counts exceeded 10^4^, showing a highly moldy state, which did not meet the requirements of GB/T 13092 (Figure 1d).

### 3.2. Regularity of VOCs during Storage

The species and relative content of VOCs during the storage of fishmeal are shown in Figure 2, and details are shown in Appendix A. There is a significant difference in the species of VOCs between domestic and imported fishmeal. During storage, a total of 315 VOCs were detected in domestic fishmeal, including heterocyclic aromatic compounds, acids, esters, ketones, aldehydes, alcohols, ethers, phenolic compounds, nitrogen-containing compounds, and hydrocarbons. A total of 194 VOCs were detected in imported fishmeal, and ethers and phenolic compounds were not detected. Among them, only 81 VOCs were detected in both domestic and imported fishmeal during storage (Figure 2a). As the freshness grades reduced, the total relative content of VOCs significantly increased. The relative content of acids increased significantly, esters and ketones fluctuated with a slight increase, and nitrogen-containing compounds increased due to the microorganisms’ decomposition during storage (Figure 2b,c). In domestic fishmeal, the relative content of heterocyclic aromatic compounds showed an increasing trend, and aldehydes and alcohols decreased. The relative content of ethers and phenolic compounds was less, showing an increasing trend (Figure 2b). In imported fishmeal, the relative content of heterocyclic aromatic compounds decreased, and aldehydes and alcohols fluctuated throughout the storage (Figure 2c).

#### 3.2.1. Heterocyclic Aromatic Compounds

Due to the thermal reactions during the process of fishmeal, accompanied by the Maillard reaction and Strecker degradation, low-molecular-weight heterocyclic aromatic compounds combined with nitrogen, oxygen, and sulfur to form new carbon rings or heterocyclic aromatic compounds. The relative content of heterocyclic aromatic compounds containing nitrogen (pyridines and pyrroles), oxygen (furans), and sulfur (thiazoles and thiophenes) increased during the later storage stage. As the reduction in freshness grade too place, the fishy smell gradually became stronger. The relative content of *2,5-dimethylpyrazine* with the odor of roasted and potato decreased during storage, decreasing to 1.33 μg/kg. Food additives added to fishmeal were detected during storage, such as *ethoxyquin*, which was commonly used in premixes, fishmeal, and fat-added products. It could be used to prevent the oxidation of vitamin A, vitamin D, and vitamin E and fat, and had anti-mold and preservation effects. To improve the palatability of feed, *1-methylnaphthalene* with the odor of coal tar or camphor ball was detected in domestic fishmeal.

#### 3.2.2. Acids and Esters

Acids are mainly formed by the hydrolysis and oxidation of fat. Saturated fatty acids, including *hexadecanoic acid*, *tetradecanoic acid*, *pentadecanoic acid*, *dodecanoic acid*, and *octadecanoic acid*, were detected, which have been speculated to be the cause of the fishy smell in the previous study [23]. *Acetic acid* and *propanoic acid* were detected in domestic fishmeal, both of which could be used as antibacterial preservatives in feed. That is, 0.6–6% added to the feed could be used to kill harmful bacteria, such as Salmonella. 

Esters are mainly formed by the esterification of alcohols with carboxylic acids decomposed liquids by microorganisms and enzymes. Most esters have the characteristic aroma of sweet and caramel [24]. Due to the Millard reaction, triglycerides, phospholipids, and fatty acids were degraded via a thermal reaction to produce lactone compounds. In the middle and later stages of storage, lactone compounds (*delta-undecanolactone* and *delta-dodecalactone*) could be detected in domestic fishmeal, showing an increasing trend.

#### 3.2.3. Ketones, Aldehydes and Alcohols

Ketones are mainly formed by the oxidation of unsaturated fatty acids or the oxidative decomposition of amino acids and the Maillard reaction, which have an enhanced effect on bloody smell [25]. Due to the Maillard reaction, ketones containing nitrogen and sulfur were detected in the later stage of storage, whose relative content showed an increasing trend. Ketenes could interact with aldehydes to enhance the fishy smell, whose relative content increased in the later stage of storage. In the meanwhile, the relative content of ketones containing aromatic function group with the function of perfume decreased.

Aldehydes are mainly formed by the decomposition of fatty acids and phospholipids, which would bring obvious smell changes due to their low odor threshold [26]. With the sour and burnt odor produced by protein corruption and oil rancidity becoming stronger during storage, the relative content of *4-ethylbenzaldehyde* with the meaty odor, *(2E,4E)-nona-2,4-dienal* with pleasant and fried fat odor, and *(2E,6Z)-nona-2,6-dienal* with the green odor of cucumber decreased in domestic fishmeal. Meanwhile, the relative content of *heptanal* with a disgusting odor increased. *Nonanal* with the odor of grass and fat was detected in the later stage of storage, showing an increasing trend. In the study of Drumm and Spanier, it was confirmed that *nonanal* was one of the main products of *oleic acid* oxidation and could be used as a marker of spoilage [27]. The relative content of *benzaldehyde* with the odor of easter and cucumber decreased during storage in imported fishmeal. *(2R,2’R,5’S)-lilac aldehyde* was detected only in the early storage stage with the odor of almonds and cloves [28,29]. 

Alcohols are mainly formed by the decomposition of fatty acids’ secondary hydroperoxides, as well as the reduction of sugar, amino acids, and carbonyl compounds. The enols with a lower odor threshold than the saturated alcohols had a significant impact on the odor during the reduction of freshness grade. *Oct-1-en-3-ol*, with a strong, sweet, mushroom-like odor, was a degradation product of linoleic acid hydroperoxide and could be used as an antibacterial agent. The relative content of *oct-1-en-3-ol* gradually decreased and could not be detected in the later stage of fishmeal storage. In the study of Igkesias et al., the relative content of *oct-1-en-3-ol* was significantly correlated with peroxide value and thiobarbiturate [30]. In the study of Jin Yang et al., the relative content of *oct-1-en-3-ol* was used to reflect the degree of rancidity of squid [31].

#### 3.2.4. Others

*Trimethylamine*, one of the nitrogen-containing compounds was detected during the fishmeal storage, with a strong ammonia and fishy odor. The relative content of *trimethylamine* increased. Due to the low odor threshold, it had a great influence on the smell changes during spoilage and usually was used as a marker of freshness. *1-(3,5-ditert-butyl-4-hydroxyphenyl)ethanone oxime* appeared in the middle storage stage, and its relative content showed an increasing trend. Under acidic conditions, amides and ketoximes would be converted and present in different forms of nitrogen-containing compounds. 

Hydrocarbons are mainly formed by the decomposition of fatty acid alkoxy radicals. *Paraffin hydrocarbons (C_6_-C_19_)* have been detected, but the odor thresholds were relatively high, and the contribution to odors was not significant. Olefins may form products with relatively low odor thresholds, such as aldehydes and ketones, under certain conditions, which were potential factors for VOCs with new odors, had a synergistic accumulation effect on VOCs, and had a significant impact on the formation of fishy odor. *2,6-ditert-butyl-4-methylphenol*, commonly used as a food additive in feed, was detected in domestic fishmeal. As an antioxidant, it could inhibit the oxidation of fat to remain fresh.

### 3.3. Correlation Analysis between VOCs and Freshness Indexes

Tests for the significance of Pearson’s correlation coefficient between freshness indexes and VOCs were adopted to analyze the correlation during fishmeal storage. The results of significant correlation are shown in Figure 3, and details are shown in Appendix A. Only nine VOCs were significantly correlated with freshness indexes in domestic and imported fishmeal. Among them, *5-pentylbenzene-1,3-diol*, *3-phenylpyridine*, and *2-hexadecyloxirane* were significantly correlated with VBN; *pentadecanoic acid* and *2,3-dihydroxypropyl (E)-octadec-9-enoate* were significantly correlated with pH value; and *hexadecanoic acid*, *tetradecanoic acid*, and *(Z)-hexadec-9-enoic acid* were significantly with both VBN and pH value. In domestic fishmeal, *hexadecanoic acid*, *tetradecanoic acid*, and *(Z)-hexadec-9-enoic acid* were significantly correlated with all freshness indexes (Figure 3a). Namely, *hexadecanoic acid* was significantly positively correlated with VBN (*p* < 0.05), and it was extremely significantly positively correlated with the other three freshness indexes (*p* < 0.01). *(Z)-hexadec-9-enoic acid* was significantly positively correlated with AV and pH value (*p* < 0.05), and it was extremely significantly positively correlated with VBN and mold counts (*p* < 0.01). *Tetradecanoic acid* was significantly positively correlated with VBN and mold counts (*p* < 0.05) and was extremely significantly positively correlated with AV and pH values (*p* < 0.01). 

VBN primarily consists of compounds such as ammonia, primary amine, secondary amine, and tertiary amine. Under storage with a high RH, it would accelerate the decomposition rate of nitrogen-containing compounds. Therefore, VBN was significantly correlated with VOCs containing nitrogen, such as *3-phenylpyridine*, *ethyl 2-benzamidoacetate*, *trimethylamine*, and so on. AV characterizes the content of carboxylic acid compounds formed by fat oxidation. Therefore, AV was significantly correlated with VOCs containing carboxyl groups, such as *pentadecanoic acid*, *dimethyl benzene-1,2-dicarboxylate*, and so on. Since the hydrocarbons are formed by the decomposition of fatty acid alkoxyl groups, AV had a more significant correlation with paraffin hydrocarbons and olefins than other freshness indexes. A Maillard reaction occurred in the process of fishmeal decomposition. The main factors affecting the Maillard reaction included reducing sugar, the types and addition content of amino acids, temperature, time, pressure, pH value, and substrate concentration. In the storage of fishmeal, the pH value fluctuated in the range of 5.8 and 6.5, which was suitable for the Maillard reaction in the study of Song Ze [32]. However, due to the relatively small change in pH value, the correlation between pH value and Maillard reaction products was small. There was a significant correlation between pH and organic acids as regulators, such as *pentadecanoic acid* and *(4Z,7Z,10Z,13Z,16Z,19Z)-docosa-4,7,10,13,16,19-hexaenoic acid*. Because the reproduction of mold needs nutrients like water, carbon sources, nitrogen sources, inorganic salt, vitamins, and others, it would accelerate the decomposition rate of VOCs containing nitrogen. Therefore, mold counts were significantly correlated with VOCs containing nitrogen, such as *azanium acetate* and *N-dodecyl-N’-(1,3-thiazol-2-yl)oxamide*. Meanwhile, molds produced organic acids during the physiological processes, which would lead to changes in the pH value. So, there was also a significant correlation between mold counts and pH regulators.

### 3.4. Determination of the Characteristic VOCs Based on PLS-DA

PLS-DA is a supervised discriminant analysis method commonly used in multivariate data analysis and pattern recognition [33,34]. The relationship between the input variable (*X*: the relative content of VOCs obtained based on HS-SPME-GC-MS) and the corresponding category label (*Y*: freshness grades of fishmeal) was modeled to find the main VOCs that best distinguish between different category samples (domestic and imported fishmeal). It can not only identify and explain the correlation between input variables and categories but also generate a prediction model for classifying new samples into appropriate categories. It can handle highly correlated and collinear input variables and can work effectively in small sample sizes. It can also determine the most discriminative variable for sample classification by calculating the importance of variables, thereby helping to explain and understand the results of classification models. To investigate the characteristic VOCs in the PLS-DA model, the variable importance in projection (VIP) was employed. The model encompassed a total of 96 samples, including two types of fishmeal, each with 16 storage time points, and three repetitions. According to the national standard GB/T 19164 and the study of Li Pei, the freshness grades were divided based on the value of VBN and AV, which was defined as follows: when the value of VBN ≤ 100 mg/100 g and AV ≤ 3 mg/g, the fishmeal was super fresh; when the VBN ≤ 130 mg/100 g and AV ≤4 mg/g, the fishmeal was superior fresh; when the VBN ≥ 160mg/100g, the fishmeal was completely corrupted; and in other cases, the fishmeal was corrupting [17,35]. Therefore, the *Y* variable in three models referred to domestic and imported fishmeal in three freshness grades (superior freshness, corrupting, and completed corruption). The *X* variables referred to the relative content of VOCs identified by HS-SPME-GC-MS.

According to the HS-SPME-GC-MS PLS-DA score plot, the first two components in the models for superior freshness, corrupting, and complete corruption grades, respectively, accounted for 97.5%, 82.5%, and 98.0% of the total variables. Furthermore, statistical and validation parameters such as accuracy, goodness of fit (R^2^), and goodness of prediction (Q^2^) were utilized to assess and compare the performance of the PLS-DA models. All three models exhibited high accuracy, R^2^, and Q^2^ values exceeding 0.90, indicating their accuracy and robustness. The PLS-DA score plot visually depicted the similarities and dissimilarities between samples. In the score plot, a greater distinction between domestic and imported fishmeal corresponded to a larger spatial separation between their respective locations, and vice versa [34]. According to the PLS-DA model score plot of the samples in Figure 4A–C, domestic and imported fishmeal were well separated from each other in the three models, indicating that the PLS-DA is an effective method for distinguishing domestic and imported fishmeal with different freshness grades by characteristic VOCs measured by HS-SPME-GC-MS.

The importance and explanatory power of each variable in classifying and identifying the freshness grades were evaluated using VIP calculations. A higher VIP value indicated a larger disparity in VOCs between the groups, emphasizing its significance in determining the freshness grades of fishmeal [36]. According to the VIP score histogram for three freshness grades, the VIP values of 7, 11, and 8 VOCs in the respective freshness grade were higher than 1, indicating that these VOCs were key characteristic compounds of domestic and imported fishmeal in the three freshness grades. Among them, the VIP values of *ethoxyquinoline*, *6,7,8,9-tetrahydro-3H-benzo[e]indole-1,2-dione*, *hexadecanoic acid*, and *heptadecane* were higher than 1 in all freshness grades; they are representative VOCs belonging to domestic and imported fishmeal during storage with different freshness grades. The marker VOCs in the superior freshness grade (VIP > 1) were *(E)-hexadec-9-enoic acid*, *2,6-ditert-butyl-4-methylphenol*, and *(Z)-octadec-13-enal*. The corrupting grade had *2,6-ditert-butyl-4-methylphenol*, *octadecanal*, *methyl (Z)-N-hydroxybenzenecarboximidate*, *(E)-hexadec-9-enoic acid*, *hexadecanal*, *tetradecanoic acid*, and *4-methoxy-6-3-(4-methoxy-6-oxopyran-2-yl)-2,4-bis(3-methylphenyl)cyclobutyl]pyran-2-one*; and the complete corruption had *tetradecanoic acid*, *octadecanal*, *[(Z)-octadec-9-enyl] acetate*, and *2,6-ditert-butyl-4-methylphenol*.

## 4. Conclusions

In this research, the VBN, AV, pH value, and mold counts were used as freshness indexes. The overall trend of each freshness index of domestic and imported fishmeal was upward and similar during storage. However, the value of freshness indexes in domestic fishmeal was larger and fluctuated greater. HS-SPME-GC-MS was used to study the VOCs of fishmeal during storage. A total of 315 VOCs were detected in domestic fishmeal, 194 VOCs were detected in imported fishmeal, and only 81 VOCs were detected in both domestic and imported fishmeal; ethers and phenolic compounds were detected only in domestic fishmeal. The total relative content of VOCs increased significantly. The relative content of acids, esters, ketones, and nitrogen-containing compounds increased. For other kinds of VOCs, there was no obvious rule of change, thus confirming that the VOCs of fishmeal are a dynamic process. Tests for the significance of Pearson’s correlation coefficient were adopted to analyze the correlation between freshness indexes and VOCs. Only nine VOCs were significantly correlated in domestic and imported fishmeal during storage. VOCs were correlated with freshness indexes. VBN can be judged by VOCs containing nitrogen. AV can be judged by VOCs containing carboxyl groups and hydrocarbons. The pH value can be judged by acids used to adjust the pH value. Mold counts can be judged by pH value regulators and VOCs containing nitrogen. Based on VBN and AV, the freshness grade of fishmeal during storage was divided into the categories of superior freshness, corrupting, and complete corruption; the PLS-DA model was used to determine the key characteristic VOCs of domestic fishmeal and imported fishmeal in three freshness grades; and the characteristic VOCs of each freshness grade were determined. HS-SPME-GC-MS combined with PLS-DA could identify and classify the freshness grade of fishmeal samples, providing a scientific basis and relevant explanations for the smell changes during fishmeal storage.

## Figures and Tables

**Figure 1 foods-13-01290-f001:**
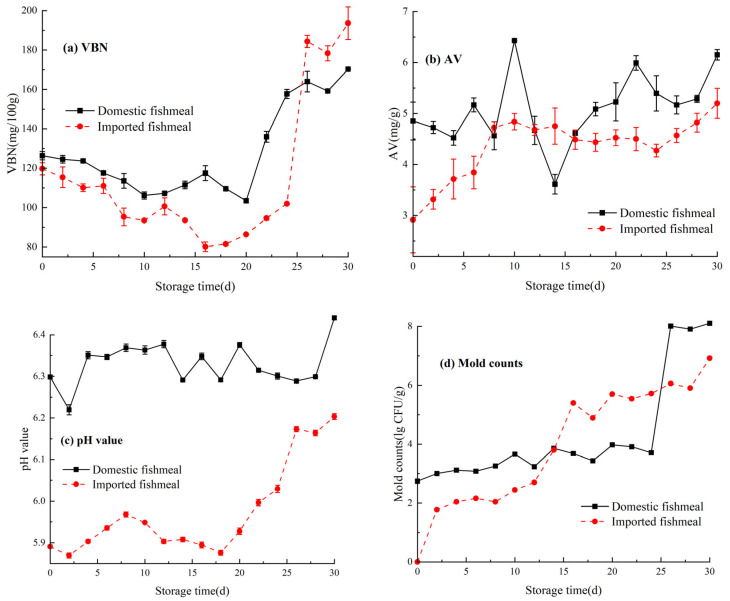
Trend of freshness indexes of fishmeal during storage: (**a**) VBN, (**b**) AV, (**c**) pH value, and (**d**) mold counts.

**Figure 2 foods-13-01290-f002:**
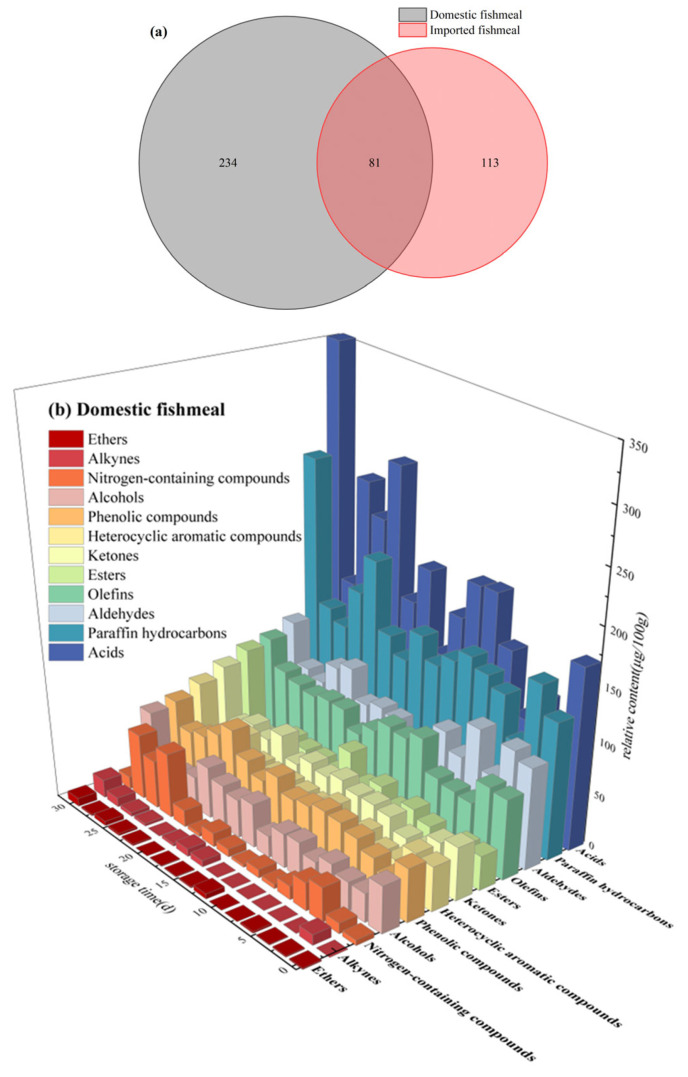
Species and relative content of VOCs during storage. (**a**) Venn diagram of species of VOCs. (**b**) Histogram of VOCs in domestic fishmeal. (**c**) Histogram of VOCs in imported fishmeal.

**Figure 3 foods-13-01290-f003:**
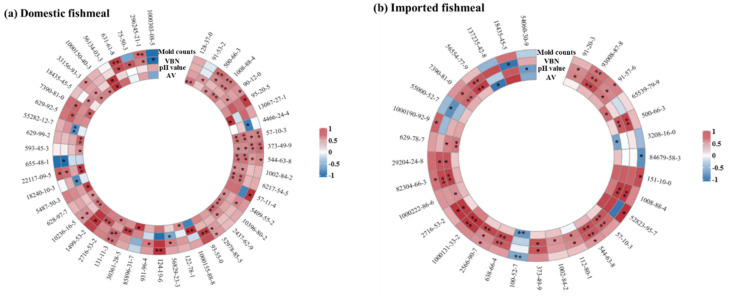
Cyclic heat map of significant correlation between each freshness index and VOCs: (**a**) domestic fishmeal and (**b**) imported fishmeal. Note: ** represents an extremely significant correlation, *p* < 0.01; * represents a significant correlation, 0.01 < *p* < 0.05. The number sequence represents the CAS number of VOCs.

**Figure 4 foods-13-01290-f004:**
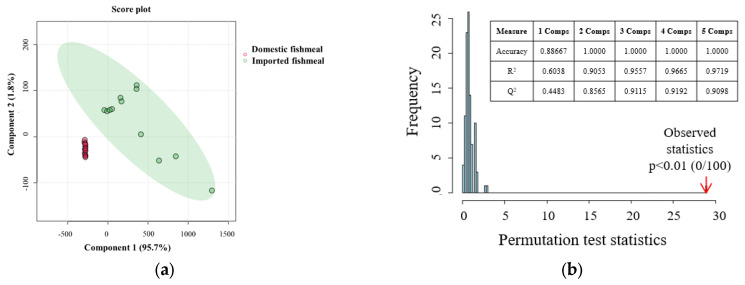
PLS-DA of HS-SPME-GC-MS data. (**A**) Superior freshness grade of domestic and imported fishmeal. (**B**) Corrupting grade of domestic and imported fishmeal. (**C**) Completed corruption grade of domestic and imported fishmeal. The figure includes (**a**) the PLS-DA score plot, (**b**) permutation test results, and (**c**) histogram of VIP scores.

**Table 1 foods-13-01290-t001:** Product labels about the composition of domestic and imported fishmeal.

Composition	Domestic Fishmeal	Imported Fishmeal
Protein	≥63%	≥68%
Fat	≤12%	≤10%
Moisture	≤10%	≤10%
Salt and sand	≤5%	≤4%
Sand only	≤2%	≤1%
Ash	≤17%	≤18%

## Data Availability

The original contributions presented in the study are included in the article; further inquiries can be directed to the corresponding author.

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
