# Peer review of "Evaluation of Dynamic Changes of Volatile Organic Components for Fishmeal during Storage by HS-SPME-GC-MS with PLS-DA"

_foods, 2024, doi:10.3390/foods13091290_

Round 1

Reviewer 1 Report

Comments and Suggestions for Authors

This study comparatively determined VOCs of domestic and imported fish meal and their change during storage. The multivariate analysis (PLS-DA) was also included. The topic, as well is interesting. However, the experimental design, which is the main point is still questioned.

Those include:

-What’s the exact difference among domestic and imported fishmeal? In my opinion, this is too wide and may not be suitable to refer to what’s different i.e compositions / processes / preparations / packaging, etc.

- The details related to each sample should provided.

- To determine the quality of fishmeal, why author focus only volatiles (as using HS-SPME-GC-MS technique), how’s bout the non-volatiles?

This detail should be added before further consideration. So, I encourage to add them and re-submit again.

Thanks

Author Response

  1. What’s the exact difference among domestic and imported fishmeal? In my opinion, this is too wide and may not be suitable to refer to what’s different i.e compositions / processes / preparations / packaging, etc. The details related to each sample should provided.

Reply: We agree with the reviewer on this point. In this work, fishmeal samples were used freshly domestic defatted fishmeal and imported Peruvian steam-dried fishmeal. The product labels about the composition of domestic and imported fishmeal are different shown in Table 1. The production process of fishmeal is: selecting raw materials, cooking, pressing, steam drying, cooling, screening, and crushing. Anchovy fish is the raw material of Peruvian fishmeal, while the raw material of domestic fishmeal is more complex with many kinds of fish. Due to the domestic defatted fishmeal, there is solid-liquid separation and oil-water separation in the production process between pressing and drying. The domestic fishmeal is usually packaged by direct filling. Considering transportation, the imported fishmeal is usually packaged in a packaging bag covered with plastic film after deodorization. We have added the details and Table 1 in Lines 84-90, 98 on Pages 2-3 of the revised paper to show the differences between domestic and imported fishmeal.

Table 1 Product labels about the composition of domestic and imported fishmeal

Composition

Domestic fishmeal

Imported fishmeal

Protein

≥63%

≥68%

Fat

≤12%

≤10%

Moisture

≤10%

≤10%

Salt and sand

≤5%

≤4%

Sand only

≤2%

≤1%

Ash

≤17%

≤18%

  1. To determine the quality of fishmeal, why author focus only volatiles (as using HS-SPME-GC-MS technique), how’s bout the non-volatiles?

Reply: Thank you for your valuable comments. We took into account freshness as the quality of fishmeal. We used the volatile base nitrogen (VBN), acid value (AV), pH value and mold counts as freshness indexes, which were not volatiles. The results of the non-volatiles are shown in Section 3.1 on Page 4 of the revised paper.

Reviewer 2 Report

Comments and Suggestions for Authors

After reviewing the manuscript entitled: “Evaluation of Dynamic Changes and Regularity of VOCs for Domestic Fishmeal and Imported Fishmeal during Storage by HS-SPME-GC-MS with PLS-DA”. This topic seems interesting. I suggest to review the document.

Abstract

The acronyms are not usually added in the abstract, review them.

Lines 9-10 indicate a PLS-DA analysis but in methodology there no was explanation of this analysis. Is that correct?

Lines 11-12, China only imports fishmeal from Peru?

Lines 21-25, the conclusion is not clear.

Introduction

Lines 51-55 yes, the application of fibers coupled to GC-MS is common in foods, it is important that the examples used will be in fish or fermented fish, including molecules indicators of fresh index. In their reference 3 determines volatile compounds of fish meal.

What is the hypothesis of this study?

Lines 73-75, review the objective of the study.

Materials and methods

Line 78-79 indicate the origin of the fishmeal, not only specifying Peruvian or domestic.

Lines 81-82 describe better fishmeal samples and how they took them.

Line 95 what is the meaning of VBN and AV.

Line 102-105 The extraction methods to obtain VOC are mixed, one is headspace and the other is using a fiber.

Line 106 what previous study?

Line 110 what is the concentration of 3-methyl-3-buten-1-ol?

Line 113-114, are the temperature and time for equilibrate and extraction of the sample?

135-139, the concentration of the internal standard there was not indicated, only at line 110 said the use of 1 µL.

Line 144. Need to review the statistical analysis. if it is possible describe a statistical model used. It is indicated that “all experiments were carried out…” but what experiments? That is not clear, because there no was explanation of the experimental design used. Later, it is specified that significant differences, but the statistical analysis was used to compare the response variables.

Line 157 maybe it the Figure 1d, because Figure 1a is VBN, and the sentence explains about microorganism load of the fish meal,

Conclusion

Review it. Seems a summary of the main results, but there was no specific conclusion of this study according to the objective.

Author Response

Reviewer 2:

  1. The acronyms are not usually added in the abstract, review them.

Reply: Thank you for pointing out the problem. We have removed the acronyms of HS-SPME-GC-MS, PLS-DA, VBN, and AV in the abstract on Page 1 of the revised paper. The acronyms of VOCs are important in the article and were repeated many times in the abstract. To make the article easy to read and concise, we used the acronyms of VOCs.

  1. Lines 9-10 indicate a PLS-DA analysis but in methodology there no was explanation of this analysis. Is that correct?

Reply: Thank you for pointing out the problem. That is correct. We referred to the structure of Reference 12, 33, and 34 published in Foods before, where the explanation of PLS-DA or OPLS-DA is in the section of results, and not in the section of methodology. Therefore, we explained PLS-DA in Section 3.4 on Page 9 of the revised paper.

  1. Lines 11-12, China only imports fishmeal from Peru?

Reply: Thank you for pointing out the problem. There are many kinds of fishmeal imported to China. Among all imported fishmeal, the Peruvian fishmeal is the most representative with the largest market share. Therefore, we used Peruvian fishmeal as imported fishmeal for comparative analysis with domestic fishmeal. In the future study, we will focus on more types of imported fishmeal.

  1. Lines 21-25, the conclusion is not clear.

Reply: Thank you for pointing out the problem. The conclusion is that based on volatile base nitrogen and acid value, the fishmeal was divided into three freshness grades, superior freshness, corrupting, and completely corrupted. By using partial least squares discriminant analysis, this study revealed the differences in flavor of the domestic and imported fishmeal during storage with different freshness grades, and it identified four common characteristic VOCs, such as ethoxyquinoline, 6,7,8,9-tetrahydro-3H-benzo[e]indole-1,2-dione, hexadecanoic acid, heptadecane produced by the fishmeal samples during storage, as well as the characteristic VOCs of fishmeal at each freshness grade. We have modified Lines 21-27 on Page 1 of the revised paper.

  1. Lines 51-55 yes, the application of fibers coupled to GC-MS is common in foods, it is important that the examples used will be in fish or fermented fish, including molecules indicators of fresh index. In their reference 3 determines volatile compounds of fish meal. What is the hypothesis of this study?

Reply: Thank you for pointing out the problem. Reference 3 is the previous study of our group. In that paper, the freshness grades of fishmeal were divided into super fresh, superior fresh, general fresh, corrupt, and completely corrupt. Five fishmeal samples with different freshness grades were selected for research. According to the previous study, the species of VOCs were determined, which changed with different freshness grades.

In the detection of freshness indexes, we found that there were differences between domestic and imported fishmeal. Therefore, we speculated whether there were differences in the species of VOCs between domestic and imported fishmeal and whether we could distinguish domestic and imported fishmeal by VOCs. Through the references, the VOCs combined with other indexes and methods could be determined as characteristics in different ripening stages, which proved that our hypothesis could be realized. Therefore, we aimed at the continuous and dynamic storage process of fishmeal with freshness degradation and determined the characteristic VOCs of domestic and imported fishmeal during storage by detecting the species and content of VOCs.

  1. Lines 73-75, review the objective of the study.

Reply: Thank you for pointing out the problem. The objective of the study is to reveal the correlations between VOCs and freshness indexes, the differences in domestic and imported fishmeal, and identify the characteristic VOCs during storage with different freshness grades. We aim to find the VOCs as freshness markers and provide a theoretical basis for storage, quality control, and subsequent utilization of fishmeal. We have clarified the objective of the study and modified it in Lines 76-78 on Page 2 of the revised paper.

  1. Line 78-79 indicate the origin of the fishmeal, not only specifying Peruvian or domestic.

Reply: Thank you for your valuable recommendation. The domestic fishmeal and imported Peruvian fishmeal were purchased from the Weihai Yongheng Seafood process plate and Technologica de Alimentos S.A. Company, respectively. We have added the information and the differences in composition, process, and package in Lines 82-91 on Page 2 of the revised paper.

  1. Lines 81-82 describe better fishmeal samples and how they took them.

Reply: Thank you for your valuable recommendation. The process of obtaining the fishmeal samples is that 5kg of fresh fishmeal laid on the enamel plate were stored in a constant temperature and relative humidity (RH) incubator at 25℃ 80%RH for 30 days respectively (RGX-250B, Tianjin Sailidesi Experimental Analysis Instrument Factory, China). According to the quartering method based on the national standard GB/T 19164, 200g fishmeal samples were taken every other day for testing. We have described the process of how we took the fishmeal samples in detail in Lines 92-96 on Page 2 of the revised paper.

  1. Line 95 what is the meaning of VBN and AV.

Reply: Thank you for pointing out the problem. The meaning of VBN is volatile base nitrogen. AV means acid value. We have added the meaning of VBN and AV in Line 110 on Page 3 of the revised paper.

  1. Line 102-105 The extraction methods to obtain VOC are mixed, one is headspace and the other is using a fiber.

Reply: Thank you for pointing out the problem. The process of extraction is that 3±0.001g of fishmeal samples, 4 mL of saturated saline, and 100 mL 3-methyl-3-buten-1-ol (1%) were added into a 20 mL headspace vial filled with silica gel, then sealed the headspace vial. After equilibrating the fishmeal samples, the 50/30 mm divinylbenzene/carboxen®/polydimethylsiloxane (DVB/CAR/PDMS) fiber head was then inserted into the vial in the headspace for 60 min at 87 ℃ to extract the VOCs. We have clarified the extraction method in Lines 127-129 on Page 3 of the revised paper.

11.Line 106 what previous study?

Reply: Thank you for pointing out the problem. According to the research of our group, we determined the optimized extraction conditions for fishmeal samples based on the single-factor and Box-Behnken design. The optimized extraction condition is as follows: 50/30 mm divinylbenzene/carboxen®/polydimethylsiloxane (DVB/CAR/PDMS) fiber head, extraction temperature of 87℃, equilibration time of 24 min, extraction time of 60 min, and the addition of a saturated saline volume of 4 mL. We have written an article about the optimized extraction condition, which is under review.

  1. Line 110 what is the concentration of 3-methyl-3-buten-1-ol?

Reply: Thank you for pointing out the problem. The concentration of  3-methyl-3-buten-1-ol is 1%. We have added the concentration in Line 124 on Page 3 of the revised paper.

  1. Line 113-114, are the temperature and time for equilibrate and extraction of the sample?

Reply: Thank you for pointing out the problem. It is correct that the temperature and time for equilibrating and extraction of fishmeal samples. We have added the object of the optimized extraction conditions in Lines 119 and 125 on Page 3 of the revised paper.

  1. 135-139, the concentration of the internal standard there was not indicated, only at line 110 said the use of 1 µL.

Reply: Thank you for pointing out the problem. The concentration of  3-methyl-3-buten-1-ol is 1%. We have added the concentration in Line 124 on Page 3 of the revised paper.

  1. Line 144. Need to review the statistical analysis. if it is possible describe a statistical model used. It is indicated that “all experiments were carried out…” but what experiments? That is not clear, because there no was explanation of the experimental design used. Later, it is specified that significant differences, but the statistical analysis was used to compare the response variables.

Reply: Thank you for pointing out the problem. The experiments are detections of freshness indexes and VOCs, which were carried out in at least three independent evaluations, expressed as the mean values ± standard deviation (SD). The analyses of variance (ANOVA) were performed among the means using the least significant difference (LSD) to analyze the differences between VOCs (P<0.05). We have clarified what experiments and specified the significant differences in the statistical analysis in Lines 161-167 on Page 4 of the revised paper.

  1. Line 157 maybe it the Figure 1d, because Figure 1a is VBN, and the sentence explains about microorganism load of the fish meal.

Reply: Thank you for pointing out the problem. It is an explanation for the increase in VBN value. We have modified the sentence in Lines 175-177 on Page 4 of the revised paper, whose expression made people confused before.

  1. Review it. Seems a summary of the main results, but there was no specific conclusion of this study according to the objective.

Reply: Thank you for your valuable comments. In the revised article, we have adjusted the Conclusion, not only a summary of results in Lines 405-429 on Page 12. The modified conclusion is as follows.

In this research, VBN, AV, pH value, and mold counts were used as freshness indexes. The overall trend of each freshness index of domestic and imported fishmeal was upward and similar during storage. However, the value of freshness indexes in domestic fishmeal was larger and fluctuated greater. HS-SPME-GC-MS was used to study the VOCs of fishmeal during storage. A total of 315 VOCs were detected in domestic fishmeal, 194 VOCs were detected in imported fishmeal, and only 81 VOCs were detected in both domestic and imported fishmeal; ethers and phenolic compounds were only detected in domestic fishmeal. The total relative content of VOCs increased significantly. The relative content of acids, esters, ketones, and nitrogen-containing compounds increased. For other kinds of VOCs, there was no obvious rule of change, which confirmed that the VOCs of fishmeal are a dynamic process. Tests for the significance of Pearson’s correlation coefficient were adopted to analyze the correlation between freshness indexes and VOCs. Only 9 VOCs were significantly correlated in domestic and imported fishmeal during storage. VOCs were correlated with freshness indexes. VBN can be judged by VOCs containing nitrogen. AV can be judged by VOCs containing carboxyl groups and hydrocarbons. The pH value can be judged by acids used to adjust the pH value. Mold counts can be judged by pH value regulators and VOCs containing nitrogen. Based on VBN and AV, the freshness grade of fishmeal during storage was determined into superior freshness, corrupting, and complete corruption, and the PLS-DA model was used to determine the key characteristic VOCs of domestic fishmeal and imported fishmeal in three freshness grades, and the characteristic VOCs of each freshness grade were determined. HS-SPME-GC-MS combined with PLS-DA could identify and classify the freshness grade of fishmeal samples, providing a scientific basis and relevant explanations for the smell changes during fishmeal storage.

Round 2

Reviewer 1 Report

Comments and Suggestions for Authors

The author has improved this manuscript as suggested before by incorporating the feedback provided earlier.

Reviewer 2 Report

Comments and Suggestions for Authors

Dear authors and Editor

All suggestions from the previous review were carried out. Only I think there no was clear for me. In the abstract, the manuscript describes different freshness grades, but later it does not describe what happens with those grades, only describing variables for domestic and imported fishmeal. is it that correct?

Author Response

  1. All suggestions from the previous review were carried out. Only I think there no was clear for me. In the abstract, the manuscript describes different freshness grades, but later it does not describe what happens with those grades, only describing variables for domestic and imported fishmeal. is it that correct?

Reply: Thank you for pointing out the problem. It is correct to refer to Reference 12. In our research, volatile base nitrogen, acid value, pH value, and mold counts are regarded as the freshness indexes. During the storage, the value of all freshness indexes of domestic and imported fishmeal increased with the freshness degradation. We have described freshness indexes change during storage in Section 3.1. Considering your valuable comments, we have added the description of the freshness indexes during storage with different freshness grades in the abstract of the revised paper.